# An Integrated Transcriptomics and Genomics Approach Detects an X/Autosome Translocation in a Female with Duchenne Muscular Dystrophy

**DOI:** 10.3390/ijms25147793

**Published:** 2024-07-16

**Authors:** Alba Segarra-Casas, Vicente A. Yépez, German Demidov, Steven Laurie, Anna Esteve-Codina, Julien Gagneur, Yolande Parkhurst, Robert Muni-Lofra, Elizabeth Harris, Chiara Marini-Bettolo, Volker Straub, Ana Töpf

**Affiliations:** 1John Walton Muscular Dystrophy Research Centre, Translational and Clinical Research Institute, Newcastle University and Newcastle Hospitals NHS Foundation Trust, Newcastle upon Tyne NE1 3BZ, UK; asegarrac@santpau.cat (A.S.-C.);; 2Genetics Department, Institut de Recerca Sant Pau (IR SANT PAU), Hospital de la Santa Creu i Sant Pau, Genetics and Microbiology Department, Universitat Autonoma de Barcelona, 08041 Barcelona, Spain; 3School of Computation, Information and Technology, Technical University of Munich, 85748 Garching, Germany; 4Universitätsklinikum Tübingen—Institut für Medizinische Genetik und angewandte Genomik, 72076 Tübingen, Germany; 5Centro Nacional de Análisis Genómico (CNAG), 08028 Barcelona, Spain; 6Universitat de Barcelona (UB), 08007 Barcelona, Spain; 7Institute of Human Genetics, School of Medicine, Technical University of Munich, 81675 Munich, Germany; 8Computational Health Center, Helmholtz Center Munich, 85764 Neuherberg, Germany; 9Muscle Immunoanalysis Unit, Newcastle upon Tyne Hospitals NHS Foundation Trust, Newcastle upon Tyne NE7 7DN, UK

**Keywords:** Duchenne muscular dystrophy, female carrier, DMD, genetic diagnosis, RNA sequencing, whole genome sequencing, translocation

## Abstract

Duchenne and Becker muscular dystrophies, caused by pathogenic variants in *DMD*, are the most common inherited neuromuscular conditions in childhood. These diseases follow an X-linked recessive inheritance pattern, and mainly males are affected. The most prevalent pathogenic variants in the *DMD* gene are copy number variants (CNVs), and most patients achieve their genetic diagnosis through Multiplex Ligation-dependent Probe Amplification (MLPA) or exome sequencing. Here, we investigated a female patient presenting with muscular dystrophy who remained genetically undiagnosed after MLPA and exome sequencing. RNA sequencing (RNAseq) from the patient’s muscle biopsy identified an 85% reduction in *DMD* expression compared to 116 muscle samples included in the cohort. A de novo balanced translocation between chromosome 17 and the X chromosome (t(X;17)(p21.1;q23.2)) disrupting the *DMD* and *BCAS3* genes was identified through trio whole genome sequencing (WGS). The combined analysis of RNAseq and WGS played a crucial role in the detection and characterisation of the disease-causing variant in this patient, who had been undiagnosed for over two decades. This case illustrates the diagnostic odyssey of female DMD patients with complex structural variants that are not detected by current panel or exome sequencing analysis.

## 1. Introduction

The *DMD* gene encodes the dystrophin protein, which binds the extracellular matrix to the cytoskeleton in the skeletal muscle fibres and is involved in sarcolemma integrity during muscle contraction [1]. Pathogenic variants in the *DMD* gene produce a weakening of the sarcolemma and muscle damage that results in muscle disease. The term dystrophinopathies includes Duchenne muscular dystrophy (DMD), Becker muscular dystrophy (BMD), and dilated cardiomyopathy (DCM) [2,3,4]. In 90% of these patients, the clinical presentation can be predicted by the reading frame rule [5,6]. DMD, the most severe phenotype with early childhood onset and rapid progression, is characterised by the absence of dystrophin expression due to out-of-frame variants. In contrast, BMD patients have a milder phenotype, later onset, and slower progression, showing reduced dystrophin staining due to a partially functional dystrophin protein caused by in-frame deletions or missense variants. 

Dystrophinopathies follow an X-linked recessive inheritance pattern, in which males are affected by the disease and female carriers are usually asymptomatic. It is estimated that 2.5–19% of female DMD carriers may manifest muscle symptoms, ranging from mild muscle weakness and myalgia to a severe DMD phenotype [7,8,9,10,11,12]. Non-muscular manifestations such as cardiomyopathy are present in 7.3–16.7% of female carriers [12]. The most common explanation for the clinical manifestation in DMD female carriers is due to a skewed X chromosome inactivation (XCI) [13,14]; however, a lack of correlation between the XCI pattern, clinical manifestations, and dystrophin alterations in muscle biopsy has been described [15].

The most prevalent pathogenic variants in the *DMD* gene are exonic deletions (60%) and duplications (10%), followed by point mutations [16,17]. However, a small number of patients (2–7%) harbour deep-intronic variants or complex chromosomal rearrangements, such as inversions, insertions, or translocations, which cannot be identified by current diagnostic techniques, such as Multiplex Ligation-dependent Probe Amplification (MLPA) and panel, or even exome, sequencing. The *DMD* gene is the largest known gene in the human genome encoding several isoforms expressed in a variety of tissues. The full-length muscle isoform Dp417m consists of 79 exons corresponding to an open reading frame of 11.3 kb [1], but 99% of the *DMD* gene consists of intronic sequences. Large introns in the *DMD* gene harbour a substantial proportion of repetitive and transposable elements (TEs) that can result in recombination events or replication errors, thus predisposing the patient to chromosomal rearrangements. This can explain the high incidence of de novo variants in the *DMD* gene, which has been estimated at 33% [17,18]. 

Here, we present a female patient with a de novo translocation between Xp21 and 17q23.2 resulting in a DMD-like phenotype. This structural variant was characterised using an integrated multi-omics approach. This study highlights the utility of transcriptome and whole genome analysis in the detection and characterisation of structural variants, in particular, in patients where clinical diagnosis might have been misguided and standard cytogenetic testing overlooked.

## 2. Results

### 2.1. Clinical Presentation

The patient was born from healthy non-consanguineous parents. Early developmental milestones were achieved at the expected times, and she achieved independent walking at 16 months old. At the age of 7 years old, she presented with a history of frequent falls and difficulty running and standing up from the floor. On examination at that time, she had prominent calf hypertrophy and pelvic girdle weakness (Medical Research Council (MRC) grade 3+ to 4/5) and got up from the floor with a Gowers manoeuvre. Her serum creatine kinase (CK) levels were 14,050 u/L. The weakness was progressive, and at 10 years old she was commenced on steroid treatment with deflazacort, which was continued until she was approximately 25 years old. She was diagnosed with osteoporosis at 21 years old. At 18 years old, the patient was able to walk very short distances holding onto furniture indoors but used a powered wheelchair outside her home. This continued until she was 24 years old when she sustained fractures to her femur and ankle, leading to a loss of independent ambulation, which resulted in a significant weight gain. 

At the most recent assessment, the patient was 29 years old and had a BMI of 39.1. She was a full-time powered wheelchair user and used a hoist for transfers. She had limited arm and leg movement with limited residual distal function preserved (MRC grade 3+ for wrists and 2 for ankle dorsiflexors). She was able to bring her right hand to her mouth with some compensation. She reported that her left arm was slightly weaker than the right one. Her spine was flexible with a slight single curvature on the back with a convexity to the right side. Her respiratory assessment showed a forced vital capacity of 0.86 L (25% predicted value), with no postural drop. Peak cough flow was also limited (208 L/min); however, an overnight pulse oximeter showed no sign of significant desaturations (Mean SPO2 98%, oxygen desaturation index for >4% of 0.7/h). Cardiac assessments have shown no evidence of impaired function, with ejection fraction >50%. Cognition was normal. 

### 2.2. Histopathological Findings

Histological analysis showed dystrophic features including fibre size variability, internalised nuclei, and severe adipose and connective tissue replacement (Figure 1A). Immunohistochemical labelling for dystrophin was absent on many fibres, with only a few small groups showing labelling (Figure 1G,H). The same groups of fibres showed an overall reduction in labelling for α, β, γ, δ sarcoglycan, and β dystroglycan (Figure 1B–F). Upregulation of utrophin was also seen (Figure 1I). This pattern of labelling was not considered typical of a manifesting DMD carrier, where a mosaic of dystrophin-positive and -negative fibres is usually observed, as the primary deficiency of dystrophin was much more severe.

Western blotting was performed alongside immunohistochemistry (Figure 2). No C-terminal dystrophin labelling could be seen but a faint, clear band was detected for the dystrophin rod domain. This band was full-sized but severely reduced in abundance in comparison to the normal control. In addition, the abundance of the sarcoglycan and dystroglycan bands was also reduced but still visible.

### 2.3. Routine Genetic Studies

Initial standard single gene screening of the *DMD* (MLPA and sequencing), *SGCA*, *SGCB*, *SCGG*, *SGCD*, *CAPN3*, *ANO5,* and *FRKP* genes was negative, and an SNP array did not detect any alteration. Later, a neuromuscular disease in silico gene panel (via whole genome sequencing) of 201 genes was negative. Structural variants (SV) were not analysed. Trio whole exome sequencing (WES) was performed in a research setting, but no candidate variant was identified amongst 429 neuromuscular-related genes (as per Töpf A. et al. 2020 [19]). In addition, the X-chromosome inactivation pattern was considered to be within the normal range (73–27%).

### 2.4. Transcriptomics and Whole Genome Sequencing

RNA sequencing (RNAseq) from muscle biopsy and trio whole genome sequencing(WGS) were performed in parallel as part of the Solve-RD project (https://solve-rd.eu/ (accessed on 15 June 2024). RNAseq data were analysed using the DROP workflow [20] within a cohort of 116 muscle samples. In brief, DROP detects genes with aberrant splicing events, aberrant expression [21], and genes with monoallelic expression. In the outlier expression analysis, the *DMD* gene was the only statistically significant outlier identified in the patient’s sample (Figure 3A), showing an 85% reduction compared to the *DMD* mean expression of all the muscle samples in the cohort (*p*-value = 1.6 × 10^−11^) (Figure 3B). In addition, this was the only sample within the cohort with the *DMD* gene as a statistically significant outlier expression. Manual inspection of the raw RNAseq data revealed an almost complete absence of reads from exon 14 to 62 of the *DMD* gene, suggesting the presence of a deletion or a structural variant encompassing the central region of the *DMD* gene. No other neuromuscular-related genes showed statistically significant outliers in the transcriptomic analysis.

Following this finding, copy number variants (CNVs) and SVs were evaluated in the trio-WGS data. No CNVs were identified within the *DMD* gene, but a putative translocation event between chromosome 17 and intron 16 of the *DMD* gene was detected in the proband. This translocation had the highest quality score and was found exclusively in this patient amongst the WGS Solve-RD cohort (n = 2303). Furthermore, this translocation was not present in the gnomAD SVs database (v4.1.0, last accession 14 June 2024). 

Raw trio-WGS data were visualised on an Integrative Genomics Viewer (IGV) and revealed in the proband the presence of chimeric reads aligning to both chromosome 17 and chromosome X (Figure 4). Allele specific PCR primers were designed to validate the exact sequence of the breakpoints and to confirm the occurrence of this de novo balanced reciprocal translocation between the *DMD* gene (chrX) and the *BCAS3* gene (chr17) (Figure 4). The breakpoints of the translocation t(X;17)(p21.1;q23.2) were chr17:58,861,454-chrX:32,578,628 (Figure 5A), and chrX:32,578,622-chr17:58,861,455 (Figure 5B), leading to two derivative chromosomes: one encoding exons 1 to 6 of the *BCAS3* gene and exons 1 to 16 of the *DMD* gene and the second one encoding exons 7 to 25 of the *BCAS3* gene and exons 17 to 79 of the *DMD* gene. A 3 bp microhomology at breakpoint 1 and an 11 bp insertion at breakpoint 2 with no homology in the nearby breakpoint sequence (300 bp) were detected. The breakpoint located in intron 16 of the *DMD* gene lies within the LINE element (L1PA10). 

Despite this translocation disrupting two protein-coding genes, *DMD* on chromosome X and *BCAS3* on chromosome 17, only *DMD* expression was affected, with normal *BCAS3* expression and splicing in RNAseq analysis.

### 2.5. Karyotype

To further confirm the translocation, a karyotype was performed on patient’s peripheral blood (Figure 6). G-band analysis shows a female karyotype with 46 chromosomes, including an apparently balanced reciprocal translocation between the short arm of a chromosome X (break at band p21) and the long arm of a chromosome 17 (break at band q23). 

## 3. Discussion

During the last decades, the increasing availability of next-generation sequencing (NGS), mainly exome and gene panel sequencing, is believed to have revolutionised the diagnostic journey of patients with rare diseases. Nonetheless, it is estimated that 50% of patients with muscle diseases remain undiagnosed after NGS testing [19,22,23]. This may be attributed to the difficulties in reclassifying variants of uncertain significance (VUS) or detecting pathogenic changes not covered by exome sequencing, such as non-coding or structural variants. In particular, RNA sequencing has been useful in the detection and interpretation of intronic cryptic variants and splicing defects [24,25,26]. In this study, we report a female patient with a DMD-like phenotype caused by a de novo reciprocal translocation t(X;17)(p21.1;q23.2) detected through a transcriptomics and genomics integrated approach. 

The proband was a female patient with childhood-onset progressive proximal weakness and elevated creatine kinase and reduced staining of the dystrophin–glycoprotein complex (DGC) on muscle biopsy (Figure 1), in addition to a reduction in abundance of DGC proteins on Western blot (Figure 2). Given the clinical presentation and histopathological findings, MLPA and sequencing of the *DMD* gene were indubitably the first line of analysis, but all proved negative. By that time, novel NGS techniques became available in the diagnostic setting and trio-WES was consequently performed but failed to identify a candidate variant, leaving the patient still undiagnosed. Years later, RNAseq and trio-WGS were performed simultaneously in a research setting. RNAseq allowed the rapid identification of *DMD* as a gene with outlier expression (Figure 3; consequently, the SVs and CNVs analysis from the trio-WGS data were focused on the *DMD* gene revealing a de novo balanced translocation t(X;17)(p21.1;q23.2) (Figure 4 and Figure 5). This translocation completely disrupts the Dp472 muscle isoform of the *DMD* gene and the *BCAS3* gene, which is an important cytoskeletal protein during embryogenesis, angiogenesis, and tumorigenesis, and has been recently identified to cause the autosomal, recessive Hengel–Maroofian–Schols syndrome [27]. 

Previous reports have proposed that the underlying mechanisms of chromosomal rearrangements in the *DMD* gene may involve a double-stranded break (DSB) followed by non-homologous end joining (NHEJ) [28], microhomology-mediated end joining (MMEJ) (also known as alternative end joining) [29], or a replication-based mechanism [30]. In the patient described here, the presence of a small microhomology (3 bp) in one of the translocation breakpoints (Figure 5) and the insertion of 11 bp not originating from the nearby sequence suggests that NHEJ, which acts on microhomologies of less than 4 bp [31], is likely to be the mechanism that caused the translocation. We hypothesise that the fact that the *DMD* breakpoint lies within a LINE element has occurred by chance, as transposable elements and repetitive sequences have been detected in chromosomal rearrangement breakpoints in a similar proportion to that found in the human genome [32,33].

The presence of clinical manifestations in female DMD carriers can be explained by (1) a *DMD* pathogenic variant in both alleles [34], (2) one *DMD* variant in the only X chromosome due to uniparental disomy or Turner syndrome (45,X) [35,36], or (3) a skewed XCI resulting in the inactivation of the wild-type allele and expression of the mutant allele, harbouring either a single nucleotide or structural variant [37]. To date, 28 balanced translocations disrupting the *DMD* gene have been reported in female dystrophinopathy patients. All the reported patients showed a DMD-like phenotype with or without cardiomyopathy and cognitive impairment and usually, but not always, present skewed XCI.

In the past, genomic rearrangements such as the one presented here were detected by karyotyping; however, small chromosomal rearrangements could not be detected, and breakpoint characterisation was not performed. The underdiagnosed prevalence of a DMD-like phenotype in female patients often leads to a misdiagnosis and/or diagnostic odyssey of female patients with muscular dystrophy and alterations in the DGC staining, as in the case described here. A secondary reduction in dystrophin expression has been described in sarcoglycanopathies, particularly in patients with pathogenic variants in β-, γ-, and δ-sarcoglycan [38]. Nevertheless, in female patients with high CK levels, a positive Gower’s sign, cardiomyopathy, and no candidate variants on exome sequencing, the presence of chromosomal rearrangements, such as deletions, insertions, inversions, or translocations within the *DMD* gene, should, be investigated by standard karyotyping or, otherwise, WGS. 

As shown here, RNAseq can frequently help in the prioritisation of WGS findings through the detection of aberrantly spliced or expressed genes, or genes with monoallelic expression. However, RNAseq alone would not have sufficed to reach the genetic diagnosis of the patient. A combination of RNAseq and WGS or, alternatively, targeted-long-read sequencing of *DMD* should be the strategy in those females with a DMD-like phenotype without a clear genetic diagnosis after routine genetic testing. Overall, this case illustrates the diagnostic odyssey in female patients with complex variants truncating the *DMD* gene that are not detected by exome or panel sequencing.

## 4. Materials and Methods

Patient. The patient was clinically assessed at the NHS England Highly Specialised Service for Rare Neuromuscular Disorders, UK. Informed consent for medical research was obtained and biological samples were submitted to the Newcastle Medical Research Council (MRC) Centre Biobank for Neuromuscular Diseases for which ethical approval was granted by the National Research Ethics Service (NRES) Committee North East–Newcastle & North Tyneside 1 (reference 19/NE/0028). 

Muscle biopsy. At the age of 9 years, a muscle biopsy from the quadriceps was taken. 

*Immunohistochemical labelling.* Labelling was performed on 6 μm sections of frozen skeletal muscle. The sections were equilibrated to room temperature after being removed from storage at −80 °C, before being permeabilized in 0.1% Triton X-100 in PBS (TBS). Samples were then incubated at 4 °C overnight in primary antibodies; dystrophin (N- and C- terminal domains), sarcoglycans (α, β, γ, and δ), dystroglycans (α and β), and utrophin. This was followed by washing in TBS. X-Cell Plus Universal probe was applied, followed by X-Cell Plus Polymer HRP, and developed with Liquid Stable DAB according to the manufacturer’s instructions. Nuclei were counterstained with Carazzi’s haematoxylin.

*Western blotting.* BioRad Protean II equipment was used to cast two 16 cm gels, 1.5 cm thick. The resolving gel was poured in two phases to give a lower phase of 7% and upper phase of 5.5% polyacrylamide. Once the resolving gel was set, a 1cm deep 3% stacking gel was poured to create the sample lanes. Frozen muscle samples were homogenised by pipette with treatment buffer. The samples were placed in boiling water for 2 min and centrifuged at 1300× *g* for 5 min. A total of 46 μL of supernatant was loaded and gels were run at 21 mA overnight at 10 °C. The gels were blotted for 7 h onto 0.45 mm nitrocellulose in transfer buffer at −13.5 °C. Immunolabelling was performed by blocking the nitrocellulose in 5% milk powder, incubating in a multiplex cocktail of primary antibodies diluted in Tris buffered saline with 0.05% Tween 20 (TBST), washing in TBST for 15 min, repeat milk blocking for 15 min, washing in TBST for 15 min, incubation in peroxidase-conjugated anti-mouse secondary antibody for 1 h, and visualisation with Supersignal West Pico in accordance with manufacturer’s instructions. 

### Genetic Studies

*DNA.* DNA from the proband and relatives was extracted from peripheral blood. MLPA of the *DMD* gene was performed using the P034 and P035 Salsa Kit (MRC-Holland, Amsterdam, Netherlands). Trio whole exome sequencing (WES) was performed at the Broad Institute of MIT and Harvard’s Genomics Platform (Cambridge, MA, USA), as described previously [39]. Single nucleotide variants (SNVs) and copy number variants (CNVs) were analysed on the *seqr* platform (seqr.broadinstitute.org). Trio whole genome sequencing (WGS) was performed as part of the Solve-RD project [40] at BGI Europe (Copenhagen, Denmark). Reads were aligned to the GRCh37 reference genome with BWA-MEM v0.7.8. The data were uploaded and analysed on the Genome-Phenome Analysis Platform (GPAP; platform.rd-connect.eu). Variants identified in trio-WES and trio-WGS were interpreted according to the American College of Medical Genetics and Genomics (ACMG) guidelines [41] and filtered by population allele frequency (<0.1%). ClinCNV and Manta v1.6.0 were used to detect CNVs and structural variants, respectively, in the trio-WGS [42,43]. Breakpoints were manually inspected using Integrative Genomics Viewer v2.16.2 (IGV). The RepeatMasker track from the UCSC Genome Browser was used to detect transposable elements or repetitive sequences nearby translocation breakpoints. Two primer pairs were used to validate the breakpoints through PCR and Sanger Sequencing: 5′ AGACCCTTTCTTTCCTGCGT 3′/5′ ACGGATGCTGGGCTCAATAT 3′ and 5′ TCAACAGCACATGTGATTTCAGTC 3′/5′ TGGGCAGCTGTAGTGAACAA 3′. 

*SNP Array.* The patient’s DNA was hybridised in a Infinium CytoSNP-850K v1.3 BeadChip array (Illumina, San Diego, CA, USA), following the manufacturer’s instructions. BlueFuse Multi v4.5 software (Illumina, San Diego, CA, USA) was used to analyse the data with an average backbone resolution of approximately 50kb and an average targeted gene resolution of approximately 10kb. Classification of the copy number variant followed the ACMG guidelines [44]. 

*RNA sequencing.* Total RNA was extracted from proband’s muscle biopsy. A Qubit RNA BR Assay kit (Thermo Fisher Scientific, Waltham, MA, USA) was used to assess the quantity and an Agilent Fragment Analyzer DNF-471 RNA Kit (15 nt) (Agilent Technologies, Santa Clara, CA, USA) was used to evaluate integrity. RNA-Seq libraries were generated from total RNA using the TruSeq Stranded mRNA Library Prep Kit (Illumina, San Diego, CA, USA). In brief, mRNA was enriched with oligo-dT magnetic beads from 500 ng of total RNA. The resulting blunt-ended, double-stranded cDNA was 3′ adenylated, and Illumina-platform-compatible adaptors with unique dual indexes and unique molecular identifiers (Integrated DNA Technologies, CA, USA) were ligated. The ligation product underwent enrichment with 15 PCR cycles. The final library was validated using an Agilent Bioanalyzer DNA 7500 assay (Agilent Technologies, Santa Clara, CA, USA). Libraries were sequenced on a NovaSeq 6000 (Illumina, San Diego, CA, USA) in paired-end mode with a read length of 2 × 151 bp, following the manufacturer’s protocol for dual indexing. Image analysis, base calling, and quality scoring were processed using the manufacturer’s software Real Time Analysis (RTA 3.4.4), followed by the generation of FASTQ sequence files. RNA-seq reads were trimmed with trim_galore/0.6.7 [45] and mapped against the hg19 human reference with STAR/2.7.8a using the - -twopassMode = BASIC parameter [46]. Mapping quality metrics were calculated with Qualimap [47], featureCounts [48], STAR log files, and custom scripts. RNAseq data were then analysed using the Detection of RNA Outliers Pipeline (DROP, v.1.3.3) [20]. This analysis was performed within a cohort of 116 muscle samples sequenced and processed using the same pipeline within the Solve-RD project. DROP was also used to verify that the RNA and DNA samples belonged to the same individual.

*X-chromosome inactivation assay.* DNA from the proband extracted from peripheral blood was subjected to an initial PCR to confirm that the patient was heterozygous for the CAG repeat in exon 1 of the androgen receptor (AR) gene using primers 5’-TCCAGAATCTGTTCCAGAGCGT (forward) and 5′-(FAM) GGCTGTGAAGGTTGCTGTTCCTCAT (reverse). Female DNA was then spiked with male DNA (2:1 ratio) to act as a control for restriction digestion. One-half of the spiked DNA was digested overnight at 37 °C with the methylation-sensitive enzymes HhaI and HpaII. The digested spiked DNA and the undigested spiked DNA were then used as templates in the AR (fluorescent) PCR. A PCR product should only be seen for an inactive (methylated) X chromosome, which cannot be digested by the enzymes, HhaI and HpaII, and can, therefore, act as a template for the PCR. The presence of the male allele in the digested sample suggests that digestion has been compromised. Following capillary electrophoresis, Genemarker (http://www.softgenetics.com/GeneMarker.html (accessed on 30 June 2023) was used to compare the peak areas of the PCR product of the digested and undigested spiked DNA and to give a percentage of X inactivation for the patient.

## Figures and Tables

**Figure 1 ijms-25-07793-f001:**
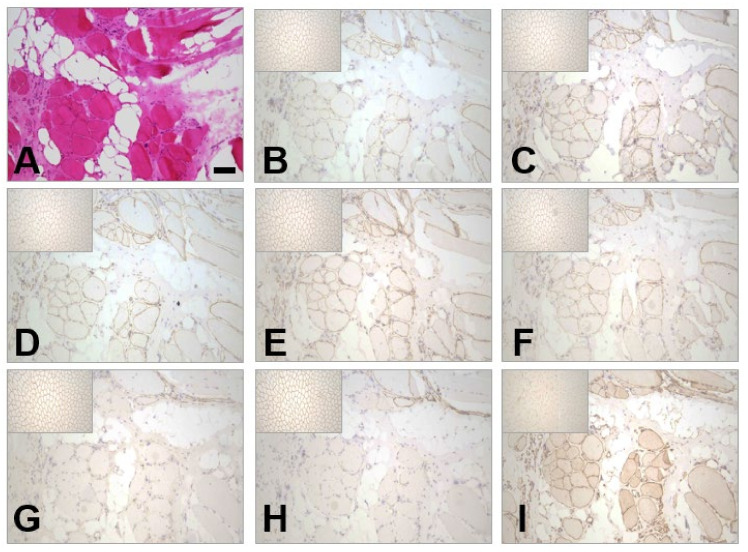
Histological analysis of patient’s skeletal muscle. (**A**) H&E-stained section shows dystrophic features: internal nuclei, clusters of basophilic, regenerating fibres, and extensive replacement of muscle fibres by fat and fibrosis (Scale bar = 100 μm). (**B**–**E**) Secondary reduction in the inmmunohistochemical labelling of the sarcoglycans: α, β, γ, and δ, respectively. (**F**) Reduction in labelling of β dystroglycan. (**G**,**H**) Labelling for N- (**G**) and C- (**H**) terminal dystrophin is severely reduced. Labelling is absent on many fibres with few remaining fibres showing reduced labelling. (**I**) Overall upregulation in utrophin labelling. All inserts in the upper left corners of the (**B**–**I**) panels show normal control immunostaining images.

**Figure 2 ijms-25-07793-f002:**
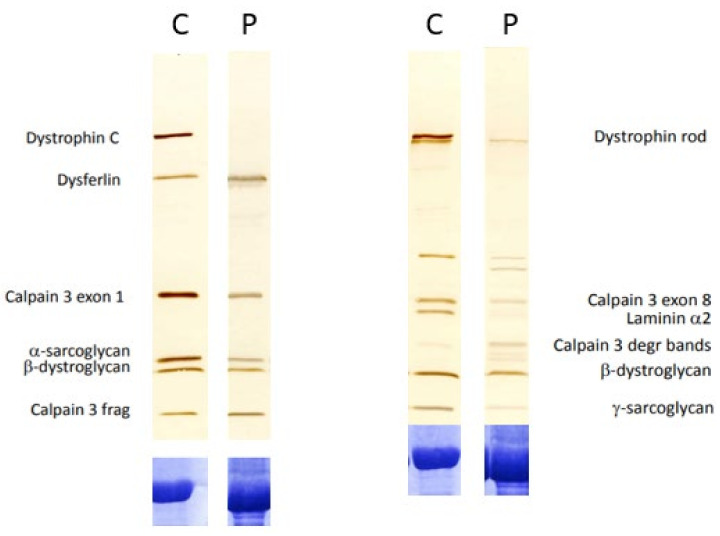
Immunoblotting analysis of normal control (C) and patient sample (P). Blots are labelled with the antibodies to the proteins indicated. In the patient, the band for C-terminal dystrophin is absent, with reductions in the abundance of α-sarcoglycan and β-dystroglycan also observed. The band for the rod domain of dystrophin is severely reduced in abundance, with the band for γ-sarcoglycan also showing a reduction in abundance.

**Figure 3 ijms-25-07793-f003:**
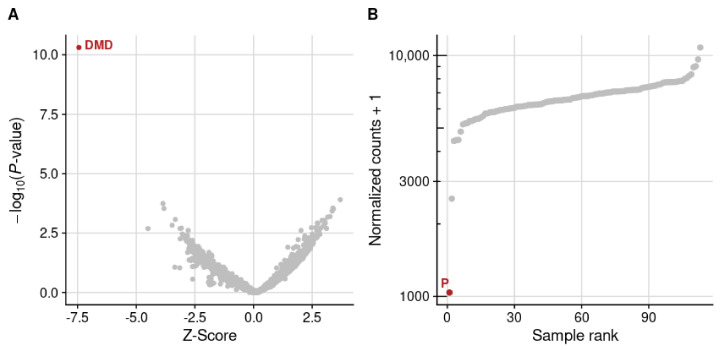
RNAseq analysis in patient’s muscle biopsy (**A**) Volcano plot from patient’s RNAseq data showing the *DMD* gene as an aberrantly expressed gene. (**B**) Expression rank plot of the *DMD* gene indicating that this patient (P) is the sample with lowest *DMD* expression. Red dots indicate statistically significant outliers detected through DROP.

**Figure 4 ijms-25-07793-f004:**
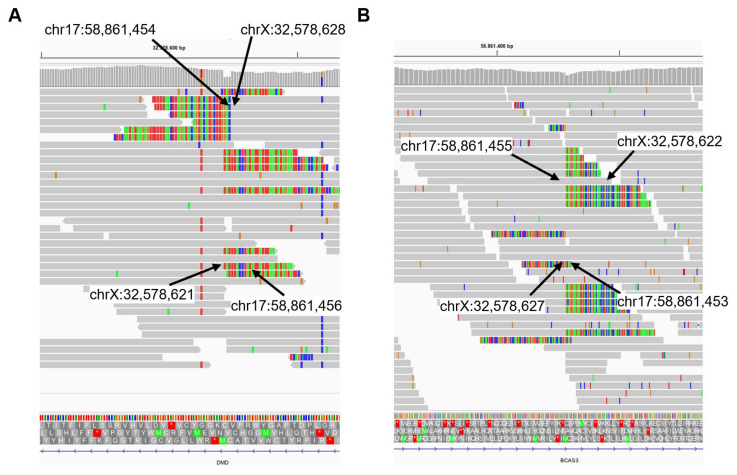
IGV visualisation of proband’s WGS showing soft-clipped reads between chromosomes 17 and X. (**A**) Translocation breakpoints in the *DMD* gene. (**B**) Translocation breakpoints in the *BCAS3* gene.

**Figure 5 ijms-25-07793-f005:**
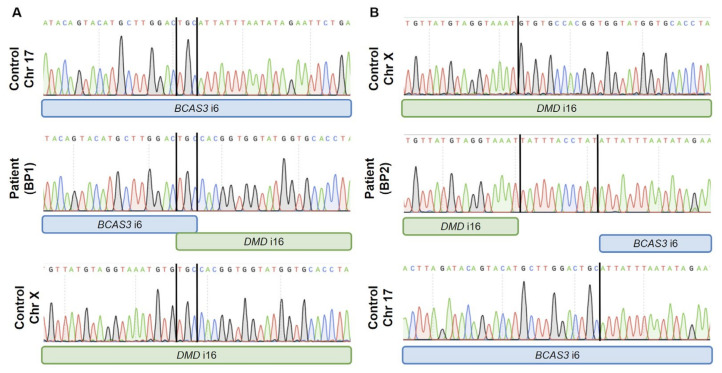
PCR validation of the translocation t(X;17)(p21.1;q23.2). Top and bottom panels show the wildtype sequence (control DNA) of the regions involved in the breakpoints in intron 6 of *BCAS3* (represented in blue) and intron 16 of *DMD* (represented in green). (**A**) Breakpoint 1 (chr17:58,861,464–chrX:32,578,628) has 3 bp microhomology between intron 6 of *BCAS3* and intron 16 of DMD. (**B**) In breakpoint 2, (chrX:32,578,622—chr17:58,861,456) an insertion of 11 bp was found.

**Figure 6 ijms-25-07793-f006:**
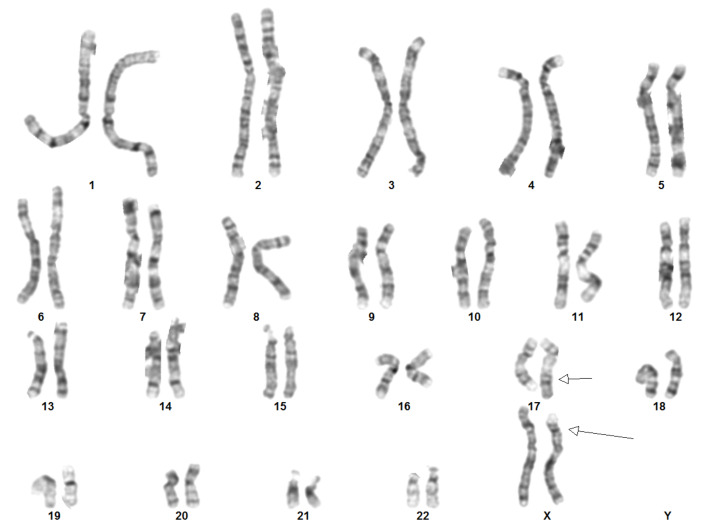
Patient’s karyotype from peripheral blood. The arrows show the translocation 46,XX,t(X;17)(p21;q23).

## Data Availability

Further details and data are available upon request from the corresponding author.

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
