# Peer review of "An Integrated Transcriptomics and Genomics Approach Detects an X/Autosome Translocation in a Female with Duchenne Muscular Dystrophy"

_ijms, 2024, doi:10.3390/ijms25147793_

Round 1
Reviewer 1 Report
Comments and Suggestions for Authors
This is an interesting work where a female patient with a DMD-like phenotype caused by a reciprocal translocation is reported. The authors describe how this translocation was detected through an integrated transcriptomics and genomics approach.
- The main question addressed by the research is the need for an approach to solve the muscular dystrophy diagnosis in cases where commonly used techniques fail.
- I consider this work is relevant and promising because they are able to find the disease causing mutation using an integrated approach. Routinely used strategies separately cannot solve complicated diagnosis and patients remain without a diagnosis during decades. They propose the required analysis combination to obtain the final diagnosis for many of these undiagnosed cases.
- The authors provide very helpful information about the possible scenarios one can find when performing molecular genetic testing. This work adds an interesting reflection on the current situation to the subject area and they provide a proposal to solve difficult diagnosis.
- I think the authors should not consider any specific improvement regarding the methodology given that they use cutting-edge methodology to overcome mutation detection problems.
- The conclusions are consistent with the evidence presented. The authors explain precisely where the problems in identifying mutations may come from and they justify perfectly the used strategy to find the mutational event. They address the posed question, they solve it and they give information to how to deal with possible difficulties in diagnostic procedures.
- Used references are appropriate.
- Regarding the figures, the supplied figures and information is adequate from my point of view. However, as a minor comment, in Figure 1, if available, I would suggest to add a control muscle image to better visualize sarcoglycans reduction and utrophin increase in the patient.
This is an excellent work that deserves its prompt publication, as it can be very helpful for other cases in which muscular dystrophy causing mutation has not been found.
Author Response
We are delighted to receive such positive feedback.
Comment: Regarding the figures, the supplied figures and information is adequate from my point of view. However, as a minor comment, in Figure 1, if available, I would suggest to add a control muscle image to better visualize sarcoglycans reduction and utrophin increase in the patient.
Reply: Thanks for this comment. Control muscle images are already shown as an insert in the upper left corner of all the immunostaining images. We have now made this clearer in the figure legend.
Reviewer 2 Report
Comments and Suggestions for Authors
In this paper, the author utilizes whole genome sequencing to detect intronic variants, which can be causative of diseases such as Duchenne Muscular Dystrophy (DMD). The discovery that combining RNA sequencing and genome sequencing enables detailed verification and diagnosis of previously difficult-to-diagnose diseases is commendable.
I would like to clarify one point. Since the expression of the DMD gene is dramatically reduced in this patient, is it not possible to diagnose DMD based on this result alone? This patient does not have exon mutations or skipping, so RNA sequencing does not reveal the cause of DMD. However, it seems that a diagnosis could still be feasible.
While this study presents a valuable case, additional examples should be included in future research to test whether DMD caused by intronic mutations can be detected with high sensitivity.
Author Response
We thank the reviewer for the positive feedback.
Comment: I would like to clarify one point. Since the expression of the DMD gene is dramatically reduced in this patient, is it not possible to diagnose DMD based on this result alone?
Reply: Thanks for bringing this up. We agree that the reduced expression of dystrophin would strongly suggest a clinical diagnosis of dystrophinopathy. While this diagnosis might suffice in some circumstances, having a confirmed genetic diagnosis is essential for accurate genetic counselling, disease management and personalised treatment.
In addition, it should be mentioned, that a reduction in the abundance of dystrophin in muscle biopsy is not exclusively an indication of dystrophinopathy. For example, secondary reduction of dystrophin levels has been reported in sarcoglycanopathies (eg Vainzof et al 1996). We have now emphasised this in the text and listed the relevant reference.